# Circadian disturbances and frailty risk in older adults

Ruixue Cai [1,2] ✉, Lei Gao[1,3,4], Chenlu Gao[1,3,4], Lei Yu[5], Xi Zheng[1], David A. Bennett[5], Aron S. Buchman[5], Kun Hu[1,3,4] & Peng Li[1,3,4] ✉

Frailty is characterized by diminished resilience to stressor events. It is associated with adverse future health outcomes and impedes healthy aging. The circadian system orchestrates ~24-h rhythms in bodily functions in synchrony with the day-night cycle, and disturbed circadian regulation plays an important role in many age-related health consequences. We investigated prospective associations of circadian disturbances with incident frailty in over 1000 older adults who had been followed annually for up to 16 years. We found that decreased rhythm strength, reduced stability, or increased variation were associated with a higher risk of incident frailty and faster progress of frailty over time. Perturbed circadian rest-activity rhythms may be an early sign or risk factor for frailty in older adults.

Frailty is defined as an age-related decline in multiple physiological systems[1]. Frail older adults have increased vulnerability to stressor events, poorer quality of life[2], and increased risks for major adverse health outcomes, including Alzheimer's disease[3–6]. Frailty has emerged as a practical and unifying concept in the care of older people who experience multi-organ problems more commonly than a single-system illness[1]. To inform appropriate interventions for preventing frailty incidence or its further progress and to promote successful aging, research is urgently needed to better understand the mechanisms of frailty. This study was designed to investigate the role of the circadian function as potential physiological correlates of frailty development.

Governed by the internal circadian clock, nearly all biological and physiological processes in humans, such as sleep and motor activity, show ~24-h rhythms as an evolutional adaptation to daily environmental changes[7]. Disrupted circadian function leads to altered rhythms in physiological processes or daily behaviors, as observed with aging and in neurodegenerative diseases[8]. For example, compared to younger adults, older people have suppressed circadian rest-activity rhythms with advanced phase[9]; and these changes were further degraded with aging within the same older adults[10]. In addition, changes in circadian rest-activity rhythms have been linked to the future development of many chronic diseases, such as type 2 diabetes, Alzheimer's disease, and Parkinson's disease[10–12]. Given the high prevalence of perturbed circadian function and frailty in older adults[13], establishing their link is important but awaits more systematic studies.

In this work, we aim to evaluate the relevance of perturbed circadian rest-activity rhythms to incident frailty and the progress of frailty over time in older adults. We hypothesize that older adults with more perturbed circadian rest-activity rhythms were at increased risk for incident frailty and had a faster deterioration in frailty progression. To test the hypotheses, we study participants in a well-phenotyped cohort of older adults: the Rush Memory and Aging Project (MAP)[14]. We employ both categorical and continuous frailty measures based on the physical frailty phenotype established in prior research[3,15]. We analyze the continuous actigraphy recordings as part of the annual follow-up in MAP for circadian rest-activity rhythms, including a set of parametric measures (i.e., amplitude and acrophase of the ~24-h component in actigraphy and the variation of cycle-to-cycle lengths) and a set of non-parametric measures (i.e., interdaily stability [IS], intradaily variability [IV], average activity level during the most active 10-h period [M10], average activity level during the least active 5-h period [L5], and relative amplitude [RA])[16–18].

[1]Medical Biodynamics Program, Division of Sleep and Circadian Disorders, Brigham and Women's Hospital, Boston, MA 02115, USA. [2]School of Public Health, Southeast University, Nanjing, Jiangsu 210000, China. [3]Division of Sleep Medicine, Harvard Medical School, Boston, MA 02115, USA. [4]Department of Anesthesia, Critical Care and Pain Medicine, Massachusetts General Hospital, Harvard Medical School, Boston, MA 02114, USA. [5]Rush Alzheimer's Disease Center, Rush University Medical Center, Chicago, IL 60612, USA. ✉e-mail: rcai1@bwh.harvard.edu; pli9@bwh.harvard.edu

## Results

### Demographics and clinical characteristics

While all alive MAP participants enrolled prior to 2005 and newly enrolled participants are eligible for the actigraphy sub-study, not everyone agreed to wear it. Compared to those who did not participate in the actigraphy assessment, participants who completed the actigraphy assessment were younger and less frail and had higher levels of education, but they did not differ by sex, vascular disease burden, and vascular risk factors. Demographics and clinical characteristics at the analytical baseline of the 1022 participants included in this current study are summarized in Table 1. The mean age at baseline was 81 years old (standard deviation: 7.2; range: 59–100 years old). Among them, 74.6% were female.

### Circadian rest-activity rhythms and incident frailty

Pairwise correlations of circadian rest-activity metrics at baseline are summarized in Supplementary Fig. 1. Over a mean of 6.6 years of follow-up, 357 (34.9% of 1022) participants developed frailty. Older age at baseline, female sex, and shorter education years were associated with increased risk for frailty. In models fully adjusted for age, sex, education, sleep duration, sleep fragmentation, vascular disease burden, and vascular disease risk, a higher risk of frailty was observed in participants with reduced amplitude, increased variation of cycle length, reduced IS, increased IV, and reduced M10. The hazard ratios (HR) of frailty corresponding to each 1-SD change in these metrics ranged from 1.21 to 1.41, with all ps < 0.05 (Table 2). The results were consistent with those obtained from models only adjusted for baseline demographics (see Supplementary Table 1). Figure 1 shows the probability of being not frail for two participants with the amplitude, variation of cycle length, IS, IV, and M10 at the 10th (low) and 90th percentile (high) in this cohort based on the fully adjusted models. Consistently, reduced 24-h amplitude (calculated using the traditional cosinor analysis) and low RA (calculated based on the nonparametric analysis) were also

### Table 1 | Demographic and clinical characteristics of participant at baseline

| Variable | Participants (n = 1022) Mean (SD) or N (%) |
|---|---|
| Age (years) | 81.0 (7.2) |
| Female | 762 (74.6%) |
| Education (years) | 15.2 (3.0) |
| Sleep duration (h) | 4.92 (1.46) |
| Sleep fragmentation (%) | 0.03 (0.01) |
| Vascular disease burden | 0.35 (0.66) |
| Vascular risk factors | 1.10 (0.80) |
| Grip strength (lbs) | 46.65 (17.58) |
| Gait speed (s) | 4.43 (1.77) |
| BMI (kg/m$^2$) | 27.25 (5.25) |
| Fatigue | 0.32 (0.55) |
| Physical activity (h) | 3.72 (3.73) |
| Frailty score | –0.11 (0.49) |
| Circadian rhythmicity characteristics | |
| Amplitude (normalized units) | 0.34 (0.10) |
| Acrophase (h) | 13.14 (1.53) |
| Variation of cycle length (h) | 1.36 (0.91) |
| IS (arbitrary units) | 0.52 (0.12) |
| IV (arbitrary units) | 1.17 (0.27) |
| M10 (counts) | 18,172.70 (10,305.50) |
| L5 (counts) | 1591.60 (1640.88) |

*SD* standard deviation, *BMI* body-mass index, *IS* interdaily stability, *IV* intradaily variability, *M10* the average activity during the most active 10-h period, *L5* the average activity during the least active 5-h period.

### Table 2 | Association of circadian rest-activity metrics and incident frailty with adjustment for covariates

| Variable | HR (95%CI) p value | HR (95%CI) p value | HR (95%CI) p value | HR (95%CI) p value | HR (95%CI) p value | HR (95%CI) p value | HR (95%CI) p value |
|---|---|---|---|---|---|---|---|
| Age[a] | 1.09 (1.07–1.11) <0.001 | 1.09 (1.07–1.12) <0.001 | 1.09 (1.07–1.12) <0.001 | 1.10 (1.08–1.12) <0.001 | 1.09 (1.07–1.11) <0.001 | 1.09 (1.07–1.11) <0.001 | 1.09 (1.07–1.11) <0.001 |
| Female sex | 3.83 (2.65–5.53) <0.001 | 3.47 (2.42–4.99) <0.001 | 3.81 (2.63–5.51) <0.001 | 3.65 (2.54–5.25) <0.001 | 3.76 (2.60–5.42) <0.001 | 3.86 (2.68–5.57) <0.001 | 3.35 (2.33–4.83) <0.001 |
| Education[b] | 1.06 (1.02–1.11) 0.004 | 1.05 (1.00–1.09) 0.030 | 1.05 (1.01–1.09) 0.020 | 1.05 (1.01–1.10) 0.012 | 1.05 (1.01–1.10) 0.012 | 1.06 (1.01–1.10) 0.008 | 1.05 (1.00–1.09) 0.029 |
| Sleep duration[a] | 1.03 (0.92–2.16) 0.558 | 1.05 (0.94–1.17) 0.403 | 1.07 (0.95–1.19) 0.255 | 1.06 (0.95–1.18) 0.333 | 1.02 (0.91–1.14) 0.766 | 1.07 (0.95–1.21) 0.283 | 1.12 (0.98–1.29) 0.085 |
| Sleep fragmentation[c] | 1.16 (1.00–1.35) 0.050 | 1.17 (1.01–1.36) 0.034 | 1.18 (1.02–1.37) 0.024 | 1.19 (1.03–1.37) 0.021 | 1.15 (0.99–1.33) 0.072 | 1.12 (0.97–1.30) 0.127 | 1.20 (1.03–1.39) 0.016 |
| Vascular disease burden | 1.14 (0.96–1.34) 0.132 | 1.22 (1.04–1.44) 0.016 | 1.16 (0.98–1.37) 0.076 | 1.17 (0.99–1.40) 0.058 | 1.19 (1.01–1.40) 0.035 | 1.15 (0.98–1.36) 0.092 | 1.21 (1.03–1.42) 0.023 |
| Vascular risk factors | 1.24 (1.07–1.44) 0.005 | 1.24 (1.07–1.45) 0.005 | 1.25 (1.07–1.45) 0.003 | 1.27 (1.09–1.47) 0.002 | 1.25 (1.08–1.45) 0.003 | 1.22 (1.05–1.41) 0.011 | 1.25 (1.07–1.45) 0.004 |
| Amplitude[d] | 1.37 (1.21–1.56) <0.001 | | | | | | |
| Acrophase[d] | | 1.04 (0.93–1.17) 0.489 | | | | | |
| Variation of cycle length[c] | | | 1.28 (1.14–1.44) <0.001 | | | | |
| IS[d] | | | | 1.27 (1.13–1.43) <0.001 | | | |
| IV[c] | | | | | 1.21 (1.07–1.38) 0.003 | | |
| M10[d] | | | | | | 1.41 (1.20–1.64) <0.001 | |
| L5[c] | | | | | | | 1.15 (0.99–1.33) 0.063 |

Cox proportional hazard regression models were used to examine the associations. Models were adjusted for age, sex, education, sleep duration, sleep fragmentation, vascular disease burden, and vascular disease risk.
*SD* standard deviation, *HR* hazard ratio, *CI* confidence interval, *IS* interdaily stability, *IV* intradaily variability, *M10* the average activity during the most active 10-h period, *L5* the average activity during the least active 5-h period.
[a]Results for 1-unit increase.
[b]Results for 1-unit decrease.
[c]Results for 1-SD increase.
[d]Results for 1-SD decrease.

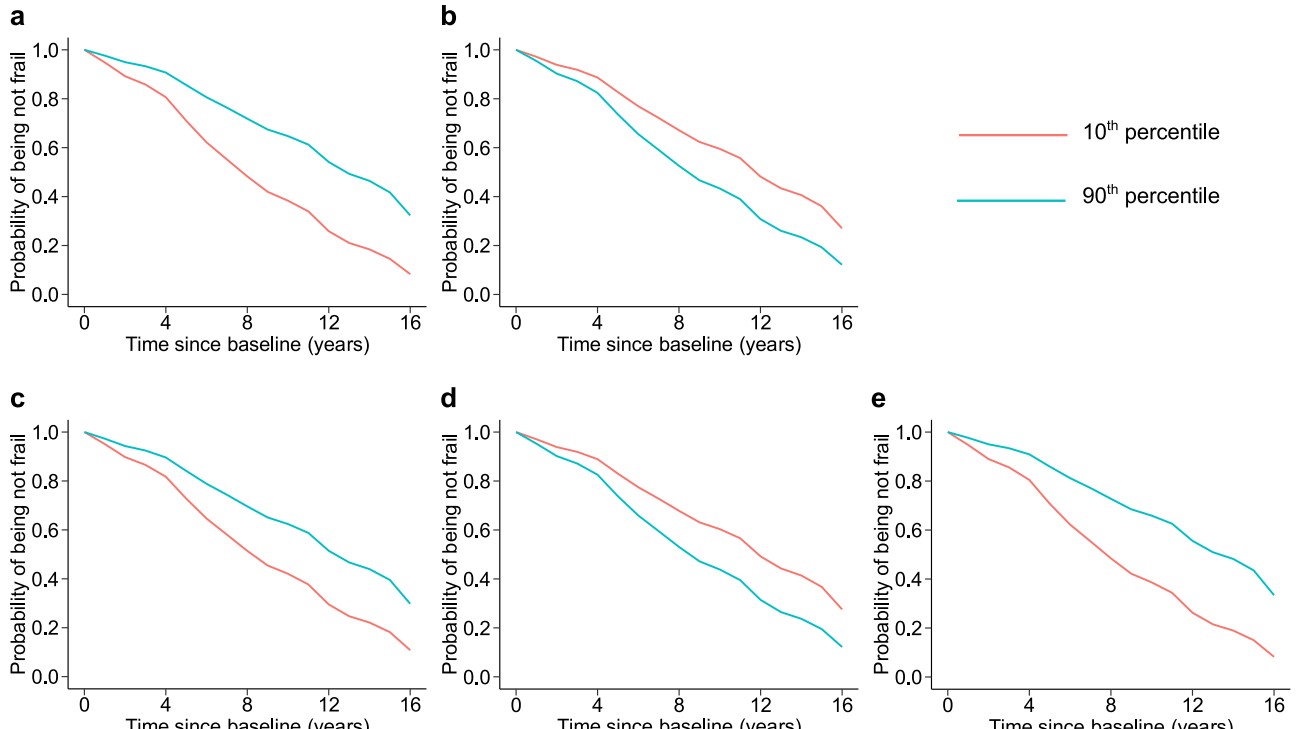

**Fig. 1 | Predicted survival curves from the fully adjusted Cox proportional hazards models.** The predicted probabilities of being not frail for two representative participants with circadian metrics (**a**) amplitude, (**b**) variation of cycle length, (**c**) IS, (**d**) IV, or (**e**) M10 at the 10th (red lines) and 90th percentiles (blue lines). IS interdaily stability, IV intradaily variability, M10 the average activity during the most active 10-h period. Source data are provided as a Source Data file.

associated with incident frailty (Supplementary Table 2). The timing of the rhythm (either acrophase calculated from UP-EMD or the 24-h acrophase from cosinor analysis) and L5 were not associated with incident frailty (Table 2, Supplementary Table 2).

The results of sensitivity analyses are summarized in Supplementary Table 3. From the competing risk regression models, the associations were consistent for amplitude, variation of cycle length, IS, and M10, but became not statistically significant for IV. When controlling for Alzheimer's and Parkinson's disease at baseline, the association of those circadian rest-activity metrics with frailty persisted, and the effect size was not substantially reduced. The results for amplitude, variation of cycle length, and IS were also retained in the models after excluding participants with baseline and incident cognitive impairment. The associations were still consistent for amplitude, variation of cycle length, IS, and M10, but became not significant for IV when we repeated the Cox models in participants with lower physical activity. For individuals under 80 years old, the associations of amplitude, IV, and M10 with the risk for frailty persisted; for individuals above 80 years old, the association of amplitude, variation of cycle length, and IS with frailty were even more pronounced, although the association of IV with frailty became not statistically significant. Therefore, the results of sensitivity analyses were consistent with the results obtained from the main analyses.

### Circadian rest-activity rhythms and change in frailty
The continuous measure for frailty increased over time with an average annual increase of 0.095 units (standard error [SE]: 0.002; $p < 0.001$), and the annual increase was accelerated by 0.002 units (SE: 0.0003) when a participant was one year older than the cohort mean at baseline ($p < 0.001$). After adjusting for age, sex, education, sleep duration, sleep fragmentation, vascular disease burden, vascular risk factors, and their interactions with time, the increase in the continuous frailty measure was much faster in participants with smaller amplitude or smaller M10, i.e., for 1-SD decrease in amplitude or M10, the

annual increase was accelerated by $0.005 \pm 0.002$ ($p = 0.030$) or $0.005 \pm 0.002$ ($p = 0.026$) (Table 3, Supplementary Table 4). These effects were equivalent to that of being 2.5 years older than the cohort mean age at baseline. These results were also consistent with results obtained from models adjusted for demographics only (see Supplementary Table 5). To illustrate the association between baseline amplitude, M10, and the rate of increase in frailty score in fully adjusted models, the predicted trajectories of the continuous composite frailty score for different levels of amplitude and M10 are shown in Fig. 2. Specifically, when compared to the high (i.e., the 90th percentile) level, participants in low level (i.e., the 10th percentile) of amplitude or M10 exhibited a steeper increase in frailty score over 16 years of follow-up.

### Circadian rest-activity rhythms and change in individual components of frailty
We used similar mixed models to test the associations of baseline circadian metrics with the rate of change in each of four frailty components (i.e., grip strength, gait speed, body mass index [BMI], and fatigue), separately (Supplementary Table 6). A more rapid decline in grip strength was associated with smaller IS, greater IV, and smaller M10 after adjusting for age, sex, education, and their interactions with time. Specifically, for 1-SD decrease in IS or M10, or 1-SD increase in IV, the annual decrease in grip strength was accelerated by $0.119 \pm 0.045$ ($p = 0.008$) or $0.107 \pm 0.045$ ($p = 0.019$), or $0.142 \pm 0.049$ ($p = 0.004$), which were equivalent to the effect of being 4–5 years older than the cohort mean at baseline (note that the average annual rate of decrease in grip strength was around 1.650, and being one year older was associated with an extra decrease of 0.024 - 0.029 per year in that three models). The association of smaller IS, and greater IV with grip strength was statistically significant after further controlling for sleep duration, sleep fragmentation, vascular disease burden, vascular risk factors, and their interactions with time. None of the circadian metrics was associated with the rate of change in gait speed. A rapid decline in

**Table 3 | Circadian rest-activity metrics and change in frailty with adjustment for covariates**

| | Estimate (SE) p value | Estimate (SE) p value | Estimate (SE) p value | Estimate (SE) p value | Estimate (SE) p value | Estimate (SE) p value | Estimate (SE) p value |
|---|---|---|---|---|---|---|---|
| Intercept | −0.118 (0.027) < 0.001 | −0.125 (0.027) < 0.001 | −0.123 (0.027) < 0.001 | −0.122 (0.027) < 0.001 | −0.121 (0.027) < 0.001 | −0.114 (0.027) < 0.001 | −0.127 (0.027) < 0.001 |
| Time | 0.089 (0.004) < 0.001 | 0.088 (0.004) < 0.001 | 0.088 (0.004) < 0.001 | 0.088 (0.004) < 0.001 | 0.088 (0.004) < 0.001 | 0.090 (0.004) < 0.001 | 0.088 (0.004) < 0.001 |
| Amplitude[a] | −0.050 (0.016) 0.002 | | | | | | |
| Amplitude × time[b] | −0.005 (0.002) 0.030 | | | | | | |
| Acrophase[a] | | 0.039 (0.015) 0.010 | | | | | |
| Acrophase × time[b] | | −0.005 (0.002) 0.022 | | | | | |
| Variation of cycle length[a] | | | 0.039 (0.015) 0.010 | | | | |
| Variation of cycle length × time[b] | | | −0.002 (0.002) 0.423 | | | | |
| IS[a] | | | | −0.034 (0.015) 0.026 | | | |
| IS × time[b] | | | | −0.002 (0.002) 0.389 | | | |
| IV[a] | | | | | 0.052 (0.016) 0.001 | | |
| IV × time[b] | | | | | 0.003 (0.002) 0.255 | | |
| M10[a] | | | | | | −0.057 (0.018) 0.001 | |
| M10 × time[b] | | | | | | −0.005 (0.002) 0.026 | |
| L5[a] | | | | | | | 0.020 (0.022) 0.351 |
| L5 × time[b] | | | | | | | 0.001 (0.003) 0.861 |

Linear mixed-effects models were used to examine the associations. Models were adjusted for age, sex, education, sleep duration, sleep fragmentation, vascular disease burden, vascular risk factors, and their interactions with time.
SD standard deviation, SE standard error, IS interdaily stability, IV intradaily variability, M10 the average activity during the most active 10-h period, L5 the average activity during the least active 5-h period.
[a]Results for 1-SD change.
[b]Results for 1-SD ×1-year change.

BMI was observed in those with smaller amplitude, smaller IS, and smaller M10 after adjusting for demographics and their interactions with time. Specifically, for each 1-SD decrease in IS, the annual decrease in BMI was accelerated by $0.037 \pm 0.015$ ($p = 0.012$), which was equivalent to an accelerated decline for being 6 years older at baseline (note that the average annual rate of decrease in BMI was 0.117, and being one year older at baseline was linked to an extra decrease of 0.006 in BMI per year); for each 1-SD decrease in amplitude or M10, the annual decrease in BMI was accelerated by $0.053 \pm 0.015$ ($p < 0.001$) or $0.043 \pm 0.015$ ($p = 0.004$). These associations remained statistically significant after further controlling for other covariates. After adjusting for demographics and their interactions with time (note that baseline age was not associated with the annual change in fatigue score), worsening of fatigue over time was much faster in subjects with smaller acrophase, smaller M10 and smaller L5, i.e., for each 1-SD decrease in acrophase, or M10, or L5, the annual increase in fatigue score was accelerated by $0.020 \pm 0.010$ ($p = 0.035$), or $0.021 \pm 0.009$ ($p = 0.028$), or $0.019 \pm 0.008$ ($p = 0.023$). These associations were not significant after further controlling for other covariates.

## Discussion

By analyzing data from a cohort of over 1000 older adults, we found that disturbances in circadian rest-activity rhythms, especially reduced rhythm strength, reduced stability, and increased variation of cycle length were associated with increased risk for frailty and faster worsening of frailty progression, in particular, the decrease in grip strength, reduction of BMI, and increasing fatigue. It is known that circadian function plays an important role in many age-related diseases, such as cardiometabolic disorders[19], delirium[20], cognitive impairment[21], and Alzheimer's disease[10]. Our results add a significant pillar to this important role of circadian function in aging by showing that it is associated with the risk of developing frailty in the future.

In the current study, the average time lag between baseline actigraphy assessments and the incidence of frailty is 6.6 years, a relatively long window with detectable changes in circadian rest-activity rhythms prior to the development of frailty, suggesting a potential of these actigraphy-based measures as early indicators of future frailty. Alternatively, the disturbances in circadian rest-activity rhythms may directly pose a risk of developing frailty in older adults. Future studies are thus warranted to untangle this potentially causal relationship, and if confirmed, strategies such as lifestyle interventions to consolidate the rest-activity rhythms may be applied as a proactive way to prevent older adults from developing frailty.

We note that the association between lower circadian amplitude and incident frailty may be explained by either reduced peak activity during daytime, increased activity nadir during nighttime, or both. Our results showed that the association of incident frailty with M10, a measure of daytime activity level, was statistically significant, whereas it was not with L5, a measure of nighttime activity level. Together with the fact that the circadian metrics were extracted from the profile of daily motor activity, it is possible that reduced daytime activity or exercise, a known risk factor of frailty[22], may confound our observations. Since physical activity is one component of the frailty phenotype and is collinear with actigraphy-derived circadian measures, it is not appropriate for us to adjust for it in statistical models. To elucidate their relationships, we performed sensitivity analyses by including only participants with relatively lower physical activity. The results showed that the observed associations of incident frailty with rest-activity rhythm measures (amplitude, variation of cycle length, IS, and M10, except for IV) remained statistically significant, suggesting that the relationships between these circadian measures and frailty are potentially independent of that of physical activity.

In a previous cross-sectional study of 105 older community-dwelling subjects[23], an association between RA and frailty was reported. However, in another cross-sectional study among 69 institutionalized

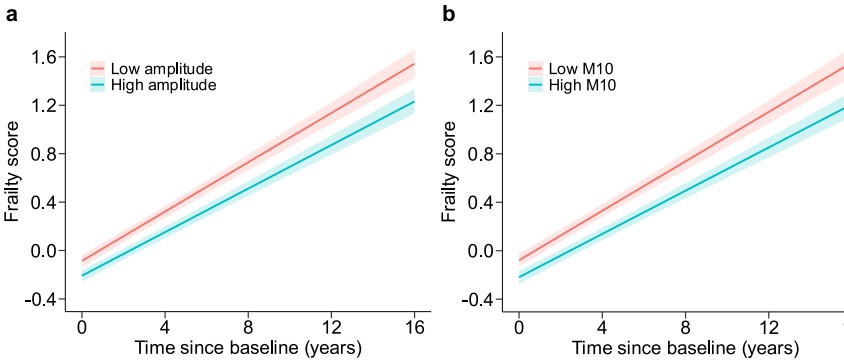

**Fig. 2 | Predicted trajectory of frailty score from the fully adjusted linear mixed-effects models.** The predicted frailty score for (**a**) amplitude or (**b**) M10 for two representative participants. The red lines represent low amplitude or M10 (10th percentile), the blue lines represent high amplitude or M10 (90th percentile), the shaded regions present the 95% confidence intervals. M10 the average activity during the most active 10-h period. Source data are provided as a Source Data file.

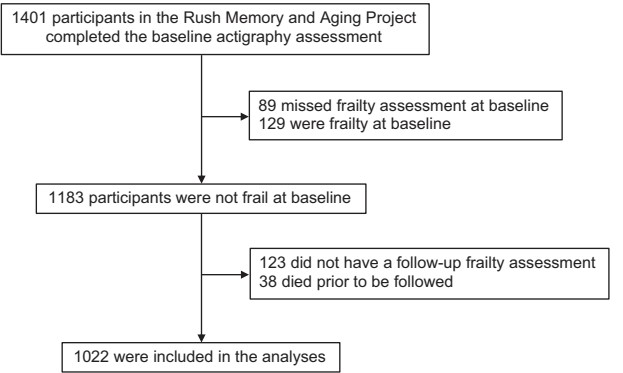

**Fig. 3 | Flow of participants through the study.** A total of 1022 participants were included in the analyses.

older adults[24], the associations between rest-activity rhythms and frailty were not significant. The discrepancy between these two studies may be attributable to differences in study population, frailty assessment, and proportion of frail participants. In addition, the cross-sectional design and the relatively small sample size in the two studies might complicate the interpretation of results. Using a longitudinal design and a much larger sample size (>1000), this study provided convincing evidence for the link between rest-activity rhythms and the development of frailty.

Previous studies have suggested that frailty is a dynamic process, i.e., within the short term, the severity of frailty may fluctuate but it tends to worsen within each individual over a longer term[3,25,26]. To document change in the degree of frailty over time, we used a continuous composite measure of frailty based on the four components[3]. We found that smaller IS, greater IV, and smaller M10 were associated with a more rapid decline in grip strength and that smaller amplitude, smaller IS, and smaller M10 were associated with a more rapid decline in BMI. Smaller acrophase, M10, and L5 were also associated with a faster increase in fatigue. Decline in strength and weight loss, mostly reflecting reduced muscle function[27] and disturbed energy balance[28], contributes to frailty. Our findings indicate that circadian disturbances may impact muscle function and energy balance more. Previous studies showed that frailty components may have different sequences of emergence before the onset of frailty, with varied results across studies[29,30]. For example, the results from the Women's Health and Aging Study II (WHAS II)[30] showed that weakness (low grip strength) tended to develop first. At the same time, weight loss and exhaustion (fatigue) predicted more rapid development of frailty. Our findings of the associations between circadian rest-activity rhythms and grip strength and BMI, may be related to the sequence in which

each component occurs and its contribution to the development of frailty. Our results demonstrate that circadian rest-activity rhythms may be sensitive to changes in individual frailty components and have the potential to be used in identifying individuals at risk for frailty at early stages.

There are many possible pathways through which circadian disturbances may lead to a higher risk of frailty. First, circadian rhythms may affect frailty through disrupted sleep. Disturbed circadian rhythms have been linked to various sleep problems or disorders, such as insomnia, poor sleep quality, and reduced sleep duration, which have been reported as predictors of frailty[31,32]. Second, disturbed circadian rest-activity rhythms have been linked previously to alterations of metabolism[11] and inflammatory markers[33] in older adults. Since metabolic function and inflammation are considered of key importance in frailty[34], circadian disturbances may contribute to an increased risk of frailty through impaired metabolism and enhanced inflammation. Finally, circadian disturbances may lead to oxidative damage and neuronal loss in the brain[35,36]. Several studies have shown that aging-related changes in the brain were associated with frailty[26,37,38]. Therefore, circadian disturbances may contribute to the development of frailty through brain aging. Growing studies suggest a possible bidirectional relationship between cognitive impairment and frailty[26,39], in which the concept of cognitive frailty was proposed[40]. Together with our prior findings, the results provide evidence that circadian disturbance may be one of the overlapped pathological mechanisms for cognitive impairment and frailty. Future studies should be conducted to elucidate the relationship between circadian rest-activity rhythms and cognitive frailty.

Note that disrupted sleep may contribute to circadian disturbances. To test whether the circadian-frailty association is independent of sleep disturbances, we included certain sleep variables derived/estimated from the actigraphy recording, i.e., sleep duration and sleep fragmentation, in the statistical models. Our results showed that the circadian-frailty association was robust, while sleep variables had no significant associations with incident frailty. However, it is worth noting that actigraphy-based sleep scoring algorithms suffer significantly from misinterpretations of sleep and wake episodes, especially when the sleep/wake diary is not available[41]. Future studies with better sleep assessment (such as combining actigraphy and sleep/wake diary and polysomnography) are warranted to formally test whether sleep disturbances are linked to frailty risk and whether the observed circadian-frailty association is mediated by or independent of sleep disturbances

Strengths of our study include the longitudinal design, long duration of follow-up, and comprehensive measures of circadian rest-activity rhythms. Our study also has several limitations. We note that the participants had a mean age of over 80 years at baseline, and only

20 were below 65 years; thus, caution should be taken when translating our findings to younger populations. In addition, rest-activity patterns can be affected by schedules and environmental conditions (e.g., light, sound, etc.) and may vary by season. Further studies on the associations between the endogenous circadian function (i.e., rhythms driven by the central clock pacemaker independent from external stimuli) and frailty are warranted. Besides, larger-scale patterns (e.g., by weeks, months, or seasons) in rest-activity rhythms and whether they contribute to better inferring future risk for frailty are also yet to be better understood.

With advances in wearable technology, actigraphy-based assessment of circadian daily rhythms can be a simple tool for long-term health monitoring in older adults. Along with other clinical and physiological measures, actigraphy-based circadian rest-activity measures may improve the identification of older adults at risk of frailty, making it possible for these individuals to benefit from early interventions (e.g., lifestyle intervention, nutritional supplementation, etc.)[42].

## Methods

### Study design
We studied participants in the Rush Memory and Aging Project (MAP)[14], an ongoing prospective, observational cohort study conducted at the Rush Alzheimer's Disease Center, Rush University Medical Center. The MAP started in 1997, and over 2100 participants have agreed to receive annual clinical evaluations, cognitive and frailty assessments. Starting in 2005, actigraphy (i.e., collection of daily motor activity data with an omnidirectional accelerometer) was added to the annual visit in the MAP. In the current study, the visit when the first actigraphy assessment was performed was used as the analytical baseline for each participant. Using the daily actigraphy data, we evaluated baseline circadian rest-activity rhythms and linked the results to incident frailty and changes in frailty during follow-up visits. The MAP was approved by an Institutional Review Board of Rush University Medical Center. All participants signed an informed consent and a repository consent to allow their data to be shared with broader research community. The current study was reviewed and approved by the Institutional Review Board of Mass General Brigham. This study followed the Strengthening the Reporting of Observational Studies in Epidemiology (STROBE) reporting guideline (Supplementary Table 7).

### Participants
A total of 1401 participants completed baseline actigraphy assessment. Among them, 218 participants were frail or did not have frailty assessment at baseline actigraphy assessment, and they were excluded. After that, 123 participants who did not have a follow-up frailty assessment and 38 individuals who died prior to be followed were further excluded. This resulted in 1022 participants who entered the subsequent analyses (Fig. 3). Follow-up of participants was until the occurrence of frailty, death, or the latest available data. By May 2022, they had been followed for up to 16 years at the time the data were frozen for these analyses (range: 1-16 years; mean: 6.6 years; standard deviation: 3.9 years).

### Assessment of circadian rest-activity rhythms
Participants wore the Actical device (Philips Respironics, Bend, OR, USA) on their non-dominant wrist for up to 14 days (range: 7–14 days). Raw data (i.e., three-dimensional accelerometer data) from the Actical device were sampled at 32 Hz and were integrated into 15-second epochs (i.e., activity counts). The activity counts recordings were subject to signal quality screenings[10]. Quality issues such as (1) isolated huge spikes with amplitude going beyond 10 standard deviations away from the individual global mean levels; and (2) sequences of zeros with duration > 60 minutes during the daytime (likely occurred when subjects took the device off) were identified and marked as gaps[43]. These gaps were treated as missing sampling points in below the signal processing for rest-activity rhythms.

Specifically, we extracted the rest-activity rhythms of ~24-h from actigraphy recordings using the uniform phase empirical mode decomposition (UP-EMD) analysis[18]. Unlike the traditional cosinor analysis, the UP-EMD analysis does not assume stationary oscillatory components in a signal (i.e., each rhythm is a sine/cosine wave with a constant amplitude and a fixed cycle length) and thus, it can better extract the biological rhythms that are oftentimes nonstationary (i.e., with varying amplitudes and cycle lengths) and nonlinear (i.e., non-sine/cosine waveforms). Using the UP-EMD-derived oscillatory component with a ~24-h cycle length, three rhythmicity measures were derived: (1) The amplitude was calculated as the absolute value of the Hilbert transform of the UP-EMD extracted ~24-h oscillatory component, which represents the strength of the rhythm. To alleviate the influence of individual activity level for a fair between-participant comparison, the amplitude was normalized by individual standard deviation of the actigraphy recording to have a normalized strength of the rhythm. (2) The acrophase which is the mean of the phase marker for peak timing across all ~24-h cycles in the oscillatory component was used to quantify the timing of peak activity level in a day. (3) The variation of cycle length was calculated as the standard deviation of all cycle lengths from the UP-EMD extracted oscillatory component. To be consistent across all participants, the above calculations were implemented to the first six complete cycles of the extracted oscillatory component, and the corresponding participant would be excluded if the oscillatory component had less than six cycles. For comparison purposes, the 24-h amplitude and acrophase of daily rest-activity rhythms were also calculated based on the traditional cosinor analysis[16].

In addition, a set of nonparametric analyses[17] were used to obtain the following rest-activity rhythm metrics: (1) Interdaily stability (IS) was used to quantify the day-to-day robustness of the 24-h activity patterns (i.e., a greater IS indicates more regular 24-h rhythm). (2) Intradaily variability (IV) was used to quantify the fragmentation of the 24-h activity patterns (i.e., a greater IV indicates more fragmented rest-activity rhythm). (3) The average activity level during the most active 10-h period (M10) was calculated to represent physical activity intensity during the active period (usually when awake; a higher M10 indicates higher activity intensity). (4) The average activity level during the least active 5-h period (L5) was calculated to represent the level of restfulness during the inactive period (usually during sleep; a lower L5 represents more restfulness). And (5) relative amplitude [RA, calculated as (M10-L5)/(M10 + L5)] was used to estimate the strength of the 24-h rest-active rhythm. RA was categorized into low and high according to the median to correct the left-skewness.

The above actigraphy analyses for rest-activity rhythms were performed using a customized software application, ezActi2, developed using MATLAB (Ver. R2022a, The MathWorks, Inc., Natick, MA, USA)[44,45].

### Assessment of frailty
Frailty was assessed annually based on five components[3,46], including grip strength, gait speed, body mass index (BMI), fatigue, and physical activity. Grip strength was measured using the Jamar hydraulic hand dynamometer (Lafayette Instruments) and was the average readout across four trials (two per hand). Gait speed was based on the time to walk eight feet. BMI was calculated as weight divided by height (kg/m²). Fatigue was assessed using two questions derived from a modified version of the Center for Epidemiologic Studies-Depression Scale (CES-D): (1) I felt that everything I did was an effort, and (2) I could not get "going". Self-reported physical activity was based on the number of hours per week that participants engaged in five types of activities: walking, gardening, calisthenics, bicycle riding, and swimming.

The scores in the lowest quintile of gait speed, grip strength, BMI, and physical activity, and a response of "yes" to one or both questions for fatigue, were considered consistent with frailty[3]. Due to level differences in performance measures between men and women, sex-specific quintiles were used for grip strength, gait speed, and physical activity. Similar to the Fried Frailty Phenotype[15], frailty was defined as the presence of 3 or more components.

In addition, a previously established and verified continuous composite measure of frailty[3,47] was also computed based on grip strength, gait speed, BMI, and fatigue. The continuous frailty score was constructed by averaging the z scores of the four components, which were calculated based on the means and standard deviations from the whole cohort at baseline. For this frailty score, a greater value means that a participant is frailer. Grip strength and BMI were multiplied by −1 so that larger values reflected poorer performance. Fatigue is constructed from responses to two specific items from the CES-D questionnaire. They were scored 1 if participants answered "yes" and 0 if otherwise. The fatigue score thus ranged from 0 to 2.

### Covariates

Several demographics (age, sex, and education)[1,48], sleep characteristics[32], as well as cardiovascular dysfunction[49] are previously reported to be associated with the risk of frailty. To test whether circadian rest-activity rhythms predict incident frailty independent of these known risk factors, we considered the following covariates at baseline: age, sex, years of education, sleep duration, sleep fragmentation, vascular disease burden, and vascular disease risk. Sleep duration was calculated by the total nighttime sleep hours based on actigraphy data. Sleep fragmentation was estimated by a prior established metric based on actigraphy that estimates the transition probabilities of rest-to-activity states[50]. A greater value represents more fragmented sleep. Vascular disease burden was estimated as the sum of self-reported claudication, stroke, heart conditions, and congestive heart failure[51]. Vascular risk was evaluated by a sum of self-reported hypertension, diabetes, and smoking history[52].

### Statistical analysis

We performed three types of statistical analyses. (1) Cox proportional hazards models were used to test the associations of disturbances in circadian rest-activity rhythms with incident frailty. The duration of follow-up was defined as time in years since baseline (for those who did not develop incident frailty) or time in years between baseline and incident frailty/death. We rounded the time to the nearest integer year to account for group tied events. In each of the primary models, one of the rhythmicity measures (amplitude, acrophase, variation of cycle length, IS, IV, M10, or L5) was included as a predictor while controlling for age, sex, and education. Fully adjusted models were performed by adding sleep duration, sleep fragmentation, vascular disease burden, and vascular disease risk as additional covariates to the primary models. As secondary analyses, we performed similar Cox proportional hazards models for 24-h amplitude, 24-h acrophase, and RA. Furthermore, we conducted a series of sensitivity analyses to examine the robustness of results: (1) performed competing risk regression models with death as a competing risk; (2) repeated the Cox proportional hazards models after further controlling for the clinical Alzheimer's disease and Parkinson's disease diagnoses at baseline; (3) repeated the Cox models by excluding participants who had cognitive impairment at baseline or developed cognitive impairment during follow-up assessments; (4) restricted the Cox models to participants with relatively lower actigraphy-derived total daily activity levels (i.e., lower than the cohort median); (5) stratified participants by age and repeated the Cox models separately for individuals under 80 years old and those aged 80 years old and above. All these models were adjusted for age, sex, education, sleep duration, sleep fragmentation, vascular disease burden, and vascular disease risk. (2) Linear mixed-effects

models were used to test the associations of circadian rest-activity rhythm measures at baseline with the longitudinal change in the continuous frailty score. In each of the primary models, the frailty score was included as a continuous dependent variable, and the time since baseline (in years) and a specific metric of circadian rest-activity rhythms, and their interaction as fixed-effect predictors. In all these models, participant and time since baseline were included as random-effects predictors. Age, sex, and education were adjusted, and items for their interactions with time since baseline were included. The potential effects of sleep duration, sleep fragmentation, vascular disease burden, vascular disease risk factor, and their interactions with time were further considered and controlled. (3) To explore whether the associations specifically exist in each component of frailty, the same linear mixed-effects models were repeated for three of the four frailty components (i.e., grip strength, gait speed, BMI), and for fatigue, which is a categorical outcome with three levels, we used generalized mixed models with a random intercept and a cumulative logit link. All statistical analyses were performed in R (version 4.1.2). Statistical significance was considered at two-tailed alpha level of 0.05.

### Reporting summary

Further information on research design is available in the Nature Portfolio Reporting Summary linked to this article.

## Data availability

The data are available under restricted access from the Rush Alzheimer's Disease Center (RADC) following the data and resource sharing policy. Access can be obtained by submitting requests through the RADC platform https://www.radc.rush.edu/requests.htm. Source data supporting our findings (Figs. 1, 2, and Supplementary Fig. 1) are provided with this paper. Source data are provided with this paper.

## Code availability

The customized software application, ezActi2, for the rest-activity rhythms analysis is available at Zenodo (https://doi.org/10.5281/zenodo.8411607) and GitHub (https://github.com/pliphd/Actigraphy). The App works on MATLAB (Ver. R2022a or newer versions). Statistical analyses were performed in R (version 4.1.2) using the following packages: survival, lme4, lmerTest, cmprsk, ggplot2. No custom code was used in the statistical analyses.

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

## Acknowledgements

This work is supported by the BrightFocus Foundation Alzheimer's Research Program A2020886S (to P.L.). L.G. is supported by NIH grant R03AG067985. K.H. is supported by the NIH grant RF1AG064312 and RF1AG059867. P.L. is also supported by a Fund to Sustain Research Excellence Program from the Brigham Research Institute. The MAP is supported by NIH grant R01AG17917.

## Author contributions

R.C. and P.L. had the idea for the study and contributed to study design. R.C. drafted the manuscript. R.C. and P.L. contributed to signal processing and statistical analyses. D.A.B. and A.S.B. contributed to the design of the Rush Memory and Aging Project and obtained data. L.G., C.G., L.Y., X.Z., D.A.B., A.S.B., K.H., and P.L. revised the manuscript for important intellectual content. All authors reviewed the manuscript and approved the final version to be published.

## Competing interests

The authors declare no competing interests.
