## [Peer Review File · Nature Communications]

Reviewers' Comments:

Reviewer #1:

Remarks to the Author:

Thank you for the opportunity to review this interesting paper, in which the authors assessed the association between baseline circadian disturbances and incident frailty in 1022 older adults followed up to 16 years. This is a relatively understudied, yet important topic. The strengths of this study include the comprehensive assessment of circadian rhythms and the long follow-up. I also think that this paper is well-written, and the statistical methods are mostly appropriate. Please consider my comments below:

- The authors considered age, sex, education, sleep, and vascular disease burden and risk as covariates. As neurodegenerative diseases/Alzheimer's disease are strongly associated with both the exposure and outcome (as mentioned in the Introduction), should they be adjusted for in the models as well?
- More information about the study population and the survival models is needed. How many individuals were lost to follow-up or died? And how were they considered in the models? Please also explain more on how the timescale was defined in the Cox models.
- Were the circadian rhythm variables assessed during follow-up as well? If there is information during follow-up, the authors may take into account changes in circadian rhythm in the models.
- How was the probability in Figure 2 constructed? Was it estimated from the Cox models adjusted for all confounders, or was it based on the Kaplan-Meier estimator?
- I would suggest presenting the fully-adjusted models in the main and putting the basic models in the supplement, as the estimates are more compelling after adjusting for sleep and diseases.
- I suggest specifying in the aim that this study is assessing "physical frailty". Incident frailty can mean different thing if it is defined by other types of frailty measure (e.g., deficit accumulation frailty index).
- Introduction: it is not clear what "frailty symptoms" mean in this context. Are they referring to the five frailty phenotype criteria? It would be helpful to clarify and give some examples when defining the meaning of frailty.

Reviewer #2:

Remarks to the Author:

Cai and colleagues used data from the Rush Memory and Aging Project to test for a prospective association between circadian parameters of the rest-activity cycle and incident frailty in people aged 59-100 years) who were followed for up to 16 years (1-16 years). The findings, if robust and reproduced, represent a significant advance in the area wherein one could determine the risk for frailty simply by investigating the actigraphy reading for one week. This is a well-written manuscript.

My major concerns about this work are as follows:

1. Physical activity patterns can be greatly affected by seasonality, and 1-2 weeks of data to interpret something that could happen after 16 years appears to be a bit of an expansive conclusion. The authors do not explain whether the accelerometry data were accompanied by a sleep diary, in the absence of which, there is a possibility of misidentifying sleep episodes, which would result in not only incorrect sleep durations but also incorrect wake durations and affect the average activity levels (e.g., if a participant was reading in bed, there is a chance it would be classified as sleep, rather than sedentary activity.). These should be well explained in the limitations. The authors also mention that frailty is a dynamic process, and the severity may fluctuate. Thus, the robustness of the relationship between two measurements which are potentially fraught with large errors, is questionable.
2. The age range includes midlife adults (59-64 years) and older adults 65+. Hence the conclusions cannot be limited to older adults.
3. The above brings up another concern about the range of follow-up. One would imagine that the follow-up period was inversely proportional to the age at the first measurement; younger people would be followed for a longer time than people who were, e.g., 95 years at the first measurement. Due to the significant range in age and only one baseline measurement, stratifying people by age (either a decade or two) rather than including the full sample in the model will make

the observations clearer.

Other suggestions:

Introduction—The authors have used numerous terms related to the circadian system, circadian control, circadian dysfunction, and perturbed circadian rest-activity rhythms. Only the last term is clear. Circadian dysfunction and circadian control can mean anything. The authors should try and keep one term and explain it clearly.

Statistical analysis—why was education included as a control variable (line 164)? Is there evidence that education can affect the independent variable-- rest-activity rhythms? It would be important to explain why particular covariates were added.

Discussion—There appears to be a lot of conjecture and reaching language, especially at the end; for e.g., the authors state that improving circadian rhythms by behavioral or light treatment can slow frailty. As the authors are well aware (they are experts in this area), 'circadian rhythms' is an all-encompassing term and should be used cautiously. Only melatonin rhythms so far have been changed by light and exercise. Whether such a change would result in a better rest-activity cycle is not known and thus is very speculative.

Reviewer #3:

Remarks to the Author:

Overall comments

This study is a secondary data analysis of an established aging cohort from the Rush Memory and Aging Project (MAP) that aims to explore the potential physiological correlates of frailty development by assessing the role of the circadian system. The study utilized comprehensive rest-activity rhythm metrics to investigate the relationship between circadian rhythm and frailty, finding that reduced rhythm strength, reduced stability, and increased variation of cycle length were associated with increased risk for frailty onset and faster frailty progression.

Notably, the study's large sample size, longitudinal cohort design, and sensitivity analyses contribute to the strength and reliability of its findings. However, there is a missed opportunity to explore the potential confounding role of cognition on the observed association between circadian function and physical frailty. Given that the MAP is a landmark study on cognitive impairment and that cognitive impairment and physical frailty often coexist in older adults, it would be informative to examine how the findings change if those with baseline and incident cognitive impairment are excluded from the analysis.

Additionally, the study found that the association between the rest-activity rhythm and frailty progression was primarily driven by a decrease in grip strength and reduction of BMI. It would be helpful for the authors to provide possible explanations for this finding.

Overall, this study provides valuable insights into the role of circadian function in frailty development and progression in older adults, while highlighting the need for further investigation into potential confounding factors and underlying mechanisms.

Specific comments

Methods

Line 71: were there systematic differences in characteristics between those who completed the actigraphy assessment and those who did not?

Lines 142-143: regarding the continuous version of the fatigue variable, need to clarify how the continuous score was created based on categorical responses to the two CES-D questions. Also, why was physical activity not included in the construction of the continuous frailty score?

Section 2.5 Covariates: the rationale behind the selection of the covariates should be stated.

Results

Lines 201-203: were predicted survival curves in Figure 2 based on the Cox model adjusting for baseline demographics?

Table 1: the measurement units of the variables should be included. The definition of 'motor function' is missing from the Methods.

Discussion

Lines 37-38: it would helpful to clarify the linkage between existing evidence on the sequence of emergency of frailty criteria and the study's finding of circadian rest-activity rhythms' association with specific frailty components (i.e., grip and BMI).

Point-by-Point Response to Reviewers' Comments

Manuscript ID: NCOMMS-23-11633-T

Title: Circadian disturbances and frailty risk in older adults: a prospective cohort study

Reviewer #1:

Thank you for the opportunity to review this interesting paper, in which the authors assessed the association between baseline circadian disturbances and incident frailty in 1022 older adults followed up to 16 years. This is a relatively understudied, yet important topic. The strengths of this study include the comprehensive assessment of circadian rhythms and the long follow-up. I also think that this paper is well-written, and the statistical methods are mostly appropriate. Please consider my comments below:

We thank the reviewer for recognizing the importance of this research topic and for their time spent and valuable comments. These suggestions are very helpful and important for us to improve the quality of this manuscript.

1. The authors considered age, sex, education, sleep, and vascular disease burden and risk as covariates. As neurodegenerative diseases/Alzheimer's disease are strongly associated with both the exposure and outcome (as mentioned in the Introduction), should they be adjusted for in the models as well?

We thank the reviewer for providing this excellent point. To address their concerns, we have performed two additional analyses: (1) we have repeated the Cox proportional hazards models after further controlling for the clinical Alzheimer's disease and Parkinson's disease diagnoses at baseline; and (2) we have also repeated these models by excluding participants who had cognitive impairment at baseline or developed cognitive impairment during follow-up assessments. The associations of circadian rest-activity metrics with incident frailty persisted. We have included the description of these sensitivity analyses and the new results in the revised manuscript. See Page 5 and Page 7 in the main text and Table S3 in Supplemental Materials.

Page 5:

"Furthermore, we conducted a series of sensitivity analyses to examine the robustness of results...2) repeated the Cox proportional hazards models after further controlling for the clinical Alzheimer's

disease and Parkinson's disease diagnoses at baseline; 3) repeated the Cox models by excluding participants who had cognitive impairment at baseline or developed cognitive impairment during follow-up assessments.”

Page 7:

“The results of sensitivity analyses are summarized in Table S3...When controlling for Alzheimer’s disease and Parkinson's disease at baseline, the association of those circadian rest-activity metrics with frailty persisted, and the effect size was not substantially reduced. The results for amplitude, variation of cycle length, and IS were also retained in the models after excluding participants with baseline and incident cognitive impairment.”

2. More information about the study population and the survival models is needed. How many individuals were lost to follow-up or died? And how were they considered in the models? Please also explain more on how the timescale was defined in the Cox models.

We thank the reviewer for these questions. In total, 123 individuals did not have a follow-up frailty assessment (i.e., lost to follow-up); 38 individuals died prior to be followed. They were excluded from the analyses. We have added these details and updated the flow chart in the revised manuscript (Page 3; Figure 1):

“...After that, 123 participants who did not have a follow-up frailty assessment and 38 individuals who died prior to be followed were further excluded.”

Figure 1. Flow of participants through the study

The final sample of 1,022 participants were included in the Cox models. During follow-up assessments, 367 participants developed incident frailty. Of those who did not develop frailty, 318 participants died. We considered both absence of frailty development and death as censoring events. The duration of follow-up was defined as time in years since baseline (for those who did not develop the incidence) or time in years between baseline and incidence/death (for those who developed the incidence or died).. We rounded the time to the nearest integer year to account for tied events. We have clarified these in the statistical analysis part (Page 3 and Page 5 in the revised manuscript):

“Follow-up of participants was until the occurrence of frailty, death, or the latest available data.”

“The duration of follow-up was defined as time in years since baseline (for those who did not develop incident frailty) or time in years between baseline and incident frailty/death. We rounded the time to the nearest integer year to account for group tied events.”

Due to the relatively high portion of deaths during follow-up, we also considered the impact of competing risk of death on our observed associations between circadian rest-activity rhythms and frailty. We performed competing risk regression models for each circadian rest-activity metric. The results were consistent for amplitude, variation of cycle length, IS, and M10, but became not statistically significant for IV. We presented these additional results in the revised manuscript (Page 7 in the revised manuscript; Page 4 in the Supplemental Materials):

“The results of sensitivity analyses are summarized in Table S3. From the competing risk regression models, the associations were consistent for amplitude, variation of cycle length, IS, and M10, but became not statistically significant for IV...”

3. Were the circadian rhythm variables assessed during follow-up as well? If there is information during follow-up, the authors may take into account changes in circadian rhythm in the models.

We thank the reviewer for the suggestion. We do have follow-up rest-activity assessments. One way to make use of the follow-up assessments is to examine the associations of changes in rest-activity metrics with frailty. The hypothesis would be that participants who had better maintained rest-activity rhythms had a slower increasing trend in frailty. To this end, we have examined whether the

longitudinal changes in circadian rhythm variables were associated with risk for frailty. We first extracted the participant-specific slope for each circadian variable, and then used the participant-specific slope as a predictor in the linear mixed effects models for frailty score.

However, the results were not the same as what we hypothesized. We found that participants who showed a faster declining in amplitude or M10 over time showed a slower increase in frailty score.

While examining the results, we believe that the results are highly masked by a floor effect in modelling the participant-specific changes in amplitude and M10, i.e., there was less room for the amplitude/M10 to reduce should they be lower enough or close to zero at baseline.

Therefore, a faster decline in these metrics appeared more possibly in those who had a greater amplitude or M10 value at baseline. This was supported by the strong linear correlation between person-specific slope of amplitude or M10 and its baseline value. This also puts constraints in putting both the baseline value and the corresponding slope in the same model (i.e., a concern of collinearity).

Therefore, we decided not to report these observations to avoid confusion. These results also suggest that caution should be exercised in interpreting participant-specific random slopes in linear mixed effects models for continuous observations with either a floor or a ceiling limit. We can include these discussions in the manuscript and additional results in Supplemental Materials if this reviewer suggest so.

4. How was the probability in Figure 2 constructed? Was it estimated from the Cox models adjusted for all confounders, or was it based on the Kaplan-Meier estimator?

The predicted probabilities were estimated from the Cox models adjusting for demographics in the initial version. We have revised Figure 2 based on the fully adjusted models in this revised version following your next suggestion (i.e., to present results from the fully adjusted models). We have made this clear in the main text and in the caption of Figure 2 (Page 6 in the revised manuscript; Figure 2):

“Figure 2 shows the probability of being not frail for two participants with the amplitude, variation of cycle length, IS, IV, and M10 at the 10th (low) and 90th percentile (high) in this cohort based on the fully adjusted models.”

5. I would suggest presenting the fully-adjusted models in the main and putting the basic models in the supplement, as the estimates are more compelling after adjusting for sleep and diseases.

Thanks for the suggestion. We have presented results from the fully adjusted models in the main text and moved results from the basic models to the supplement.

6. I suggest specifying in the aim that this study is assessing “physical frailty”. Incident frailty can mean different thing if it is defined by other types of frailty measure (e.g., deficit accumulation frailty index).

We agreed with the reviewer. We have clarified our frailty definition in the aim (Page 2 in the revised manuscript):

“We employed both categorical and continuous frailty measures based on the physical frailty phenotype established in prior research.^{3,14}”

7. Introduction: it is not clear what “frailty symptoms” mean in this context. Are they referring to the five frailty phenotype criteria? It would be helpful to clarify and give some examples when defining the meaning of frailty.

Yes, they were referring to the frailty phenotype criteria. To better clarify this, we have instead changed to use the categorical or the continuous measure of frailty based on frailty phenotype criteria.

Reviewer #2:

Cai and colleagues used data from the Rush Memory and Aging Project to test for a prospective association between circadian parameters of the rest-activity cycle and incident frailty in people aged 59-100 years) who were followed for up to 16 years (1-16 years). The findings, if robust and reproduced, represent a significant advance in the area wherein one could determine the risk for frailty simply by investigating the actigraphy reading for one week. This is a well-written manuscript.

We thank the reviewer for the positive comments and helpful feedback that have helped us strengthen the manuscript significantly.

My major concerns about this work are as follows:

1. Physical activity patterns can be greatly affected by seasonality, and 1-2 weeks of data to interpret something that could happen after 16 years appears to be a bit of an expansive conclusion. The authors do not explain whether the accelerometry data were accompanied by a sleep diary, in the absence of which, there is a possibility of misidentifying sleep episodes, which would result in not only incorrect sleep durations but also incorrect wake durations and affect the average activity levels (e.g., if a participant was reading in bed, there is a chance it would be classified as sleep, rather than sedentary activity.). These should be well explained in the limitations. The authors also mention that frailty is a dynamic process, and the severity may fluctuate. Thus, the robustness of the relationship between two measurements which are potentially fraught with large errors, is questionable.

We agree with the reviewer that physical activity patterns may have seasonal variations. This is indeed an important topic to investigate. Unfortunately, there was only one actigraphy recording for each participant during their scheduled annual follow-up in the Rush Memory and Aging Project that limits the capability to rigorously examine a seasonal effect that requires a study design with multiple assessments of physical activity across a year. As recommended by the reviewer, we have included the potential influence of seasonal patterns as a limitation and future research direction in the revised manuscript (Page 11):

“In addition, rest-activity patterns can be affected by schedules and environmental conditions (e.g., light, sound, etc.) and may vary by season. Further studies on the associations between the endogenous circadian function (i.e., rhythms driven by the central clock pacemaker that is independent from external stimuli) and frailty are warranted. Besides, larger-scale patterns (e.g., by weeks, months, or

seasons) in rest-activity rhythms and whether they contribute to better infer future risk for frailty are also yet to be better understood.”

We agree with the reviewer that sleep can be misinterpreted solely from actigraphy data. For example, sedentary or inactive time can easily be interpreted as sleep episodes from actigraphy-based sleep scoring algorithms¹. We note that the current study is focused on the association of circadian disturbances and risk for frailty. Thus, the primary actigraphy-derived measures in this study are rest-activity rhythms (instead of sleep) that have been widely accepted in population-based research to assess the behavioral rhythms or patterns.²⁻⁵ On the other hand, disrupted sleep may contribute to circadian disturbances. To test whether the circadian-frailty association is independent of the effects of sleep disturbances, we included certain sleep variables derived/estimated from the actigraphy recoding in the statistical models, i.e., sleep duration and sleep fragmentation. We acknowledge that these estimated sleep variables may be not accurate. Thus, better sleep assessment such as combining actigraphy and sleep/wake diary and PSG should be used in future studies to formally test whether sleep disturbances are linked to frailty risk and whether the observed circadian-frailty association is mediated by or independent of sleep disturbances. We have included these in the revised discussion (Page 11 in the revised manuscript):

“Note that disrupted sleep may contribute to circadian disturbances. To test whether the circadian-frailty association is independent of sleep disturbances, we included certain sleep variables derived/estimated from the actigraphy recoding, i.e., sleep duration and sleep fragmentation in the statistical models. Our results showed that the circadian-frailty association was robust while sleep variables had no significant associations with incident frailty. However, it is worth noting that actigraphy-based sleep scoring algorithms suffer significantly from misinterpretations of sleep and wake episodes, especially when the sleep/wake diary is not available.⁴⁹ Future studies with better sleep assessment (such as combining actigraphy and sleep/wake diary and polysomnography) are warranted to formally test whether sleep disturbances are linked to frailty risk and whether the observed circadian-frailty association is mediated by or independent of sleep disturbances.”

For the concern related to the dynamic nature of frailty, we considered both the incidence of frailty and the longitudinal changes of frailty symptoms and severity over time. We believe that the modelling of the longitudinal change in frailty helped address the issue that severity of frailty can fluctuate over time

and studying only the incidence of frailty can be fraught as the symptoms may still improve or worsen after the first incidence.

We also examined the association of longitudinal change in rest-activity rhythms and frailty. See our response to point 3 of Reviewer 1.

2. The age range includes midlife adults (59-64 years) and older adults 65+. Hence the conclusions cannot be limited to older adults.

Thanks for pointing this out. While further checking the data set in our study, we found that only 20 participants (2%) were aged below 65 years, and 98% participants were 65+ years. Considering this lack of representation in midlife adults in this cohort, we believe that the translation of findings to other age groups is still challenging. We have better clarified this in the revised manuscript (Page 11):

“We note that the participants had a mean age of more than 80 years at baseline, and only 20 of them were aged below 65 years; thus, caution should be taken when translating our findings to younger populations.”

3. The above brings up another concern about the range of follow-up. One would imagine that the follow-up period was inversely proportional to the age at the first measurement; younger people would be followed for a longer time than people who were, e.g., 95 years at the first measurement. Due to the significant range in age and only one baseline measurement, stratifying people by age (either a decade or two) rather than including the full sample in the model will make the observations clearer.

This is indeed a very important point. In the revised manuscript, we have performed additional sensitivity analyses to stratify the sample by age (i.e., under 80 years—the cohort mean and above 80 years) and re-investigated the associations between rest-activity rhythms and risk of frailty. The observations were highly consistent with the primary models, except that the association between IV and frailty became not statistically significant in the “80 years and above” stratum. The results were presented as part of sensitivity analyses in the revised manuscript and Table S3 (page 7 in the revised manuscript; page 4 in Supplement):

“The results of sensitivity analyses are summarized in Table S3... For individuals under 80 years old, the associations of amplitude, IV, and M10 with the risk for frailty persisted; for individuals above 80 years old, the association of amplitude, variation of cycle length, and IS with frailty were even more pronounced although the association of IV with frailty became not statistically significant.”

Other suggestions:

Introduction—The authors have used numerous terms related to the circadian system, circadian control, circadian dysfunction, and perturbed circadian rest-activity rhythms. Only the last term is clear. Circadian dysfunction and circadian control can mean anything. The authors should try and keep one term and explain it clearly.

We thank the reviewer for the suggestion. We have revised the manuscript throughout, eliminating unnecessary different terms. Specifically, we use only “circadian rest-activity rhythms” in the Introduction and Discussions sections when referring to the measures extracted from actigraphy. Only when we discuss the mechanisms underlying or interpret [perturbed] circadian rest-activity rhythms, we use the term “circadian function”.

Statistical analysis—why was education included as a control variable (line 164)? Is there evidence that education can affect the independent variable-- rest-activity rhythms? It would be important to explain why particular covariates were added.

We apologize that this is not clear in our manuscript. Education was reported to be associated with frailty in many previous studies.^{6,7} We have better clarified this and cited the corresponding literatures in the revised manuscript (Page 5):

“Several demographics (age, sex, and education)^{1,23}, sleep characteristics²⁴, as well as cardiovascular dysfunction²⁵ are previously reported to be associated with the risk of frailty. To test whether circadian rest-activity rhythms predict incident frailty independent of these known risk factors, we considered the following covariates at baseline...”

Discussion—There appears to be a lot of conjecture and reaching language, especially at the end; for e.g., the authors state that improving circadian rhythms by behavioral or light treatment can slow frailty. As the authors are well aware (they are experts in this area), ‘circadian rhythms’ is an all-encompassing term and should be used cautiously. Only melatonin rhythms so far have been changed by light and exercise. Whether such a change would result in a better rest-activity cycle is not known and thus is very speculative.

Apologies for the lack of rigor. In the revised manuscript, we have reworded these sentences and deleted the overreaching languages that appeared to be speculative.

Reviewer #3:

Overall comments

This study is a secondary data analysis of an established aging cohort from the Rush Memory and Aging Project (MAP) that aims to explore the potential physiological correlates of frailty development by assessing the role of the circadian system. The study utilized comprehensive rest-activity rhythm metrics to investigate the relationship between circadian rhythm and frailty, finding that reduced rhythm strength, reduced stability, and increased variation of cycle length were associated with increased risk for frailty onset and faster frailty progression.

We thank the reviewer for the positive comments and valuable suggestions.

Notably, the study's large sample size, longitudinal cohort design, and sensitivity analyses contribute to the strength and reliability of its findings. However, there is a missed opportunity to explore the potential confounding role of cognition on the observed association between circadian function and physical frailty. Given that the MAP is a landmark study on cognitive impairment and that cognitive impairment and physical frailty often coexist in older adults, it would be informative to examine how the findings change if those with baseline and incident cognitive impairment are excluded from the

analysis.

To address their concerns, we have performed two additional analyses: (1) we have repeated the Cox proportional hazards models after further controlling for the clinical Alzheimer's disease and Parkinson's disease diagnoses at baseline; and (2) we have also repeated these models by excluding participants who had cognitive impairment at baseline or developed cognitive impairment during follow-up assessments. The associations of circadian rest-activity metrics with incident frailty persisted. We have included these sensitivity analyses and results in the revised manuscript. See Page 5 and Page 7 in the main text and Table S3 in Supplemental Materials.

Page 5:

“Furthermore, we conducted a series of sensitivity analyses to examine the robustness of results...2) repeated the Cox proportional hazards models after further controlling for the clinical Alzheimer's disease and Parkinson's disease diagnoses at baseline; 3) repeated the Cox models by excluding participants who had cognitive impairment at baseline or developed cognitive impairment during follow-up assessments.”

Page 7:

“The results of sensitivity analyses are summarized in Table S3...When controlling for Alzheimer's disease and Parkinson's disease at baseline, the association of those circadian rest-activity metrics with frailty persisted, and the effect size was not substantially reduced. The results for amplitude, variation of cycle length, and IS were also retained in the models after excluding participants with baseline and incident cognitive impairment.”

Additionally, the study found that the association between the rest-activity rhythm and frailty progression was primarily driven by a decrease in grip strength and reduction of BMI. It would be helpful for the authors to provide possible explanations for this finding.

We thank the reviewer for the great suggestion. One possible explanation for this finding is that changes in different frailty components may be interrelated with each other. Decline in strength and weight loss, mostly reflect the reduced muscle function, and disturbed energy balance, contributing to

frailty, while circadian disturbances may have more impacts on muscle function and energy balance. We have added the explanation for our finding in the discussion (page 10 in the revised manuscript):

“Decline in strength and weight loss, mostly reflecting the reduced muscle function³⁶ and disturbed energy balance³⁷, contributes to frailty. Our findings indicate that circadian disturbances may have more impacts on muscle function and energy balance.”

Overall, this study provides valuable insights into the role of circadian function in frailty development and progression in older adults, while highlighting the need for further investigation into potential confounding factors and underlying mechanisms.

Again, we thank the reviewer for their positive comments and for recognizing the importance of this study.

Specific comments

Methods

Line 71: were there systematic differences in characteristics between those who completed the actigraphy assessment and those who did not?

Everyone in MAP who was alive when actigraphy study started were eligible to be enrolled, but some participants withdrew or refused to wear the device. We compared the age, sex, education, continuous frailty score, vascular disease burden, and vascular risk factors of the participants who completed the actigraphy assessment with those who did not complete the assessment. We found there were no significant differences in age, vascular disease burden, and vascular risk factors, but sex, education, and the continuous frailty score differed between these two groups. Participants who completed the actigraphy assessment were younger and had longer education years and smaller frailty scores. See Page 6 in the revised manuscript:

“While all alive MAP participants enrolled prior to 2005 and new enrolled participants are eligible for the actigraphy sub-study, not everyone agreed to wear it. Compared to those who did not participate in the actigraphy assessment, participants who completed the actigraphy assessment were younger and less frail, and had higher levels of education, but they did not differ with respect to sex, vascular disease burden, and vascular risk factors.”

Lines 142-143: regarding the continuous version of the fatigue variable, need to clarify how the continuous score was created based on categorical responses to the two CES-D questions. Also, why was physical activity not included in the construction of the continuous frailty score?

We apologize that this is not clear in our manuscript. We have added a sentence to clarify the construction of the continuous score for frailty in the revised methods (Page 5):

“Fatigue is constructed from responses to two specific items from the CES-D questionnaire. They were scored 1 if participants answered “yes” and 0 if otherwise. The fatigue score thus ranged from 0 to 2.”

Physical activity was not included in the construction of the MAP continuous frailty score mainly due to certain historical reason. The MAP continuous frailty score was originally introduced to examine the relationship of daily physical activity and frailty. Since then, this continuous measure has been used in many published studies using the MAP, showing the associations of this continuous frailty measure with a wide range of adverse health outcomes including mortality, incident disability, incident AD and cognitive decline.⁸⁻¹¹ It has also been used to document the rate of change in physical frailty.^{8,10} To be consistent and for a better comparability with prior MAP studies, we opt to use this established version of continuous frailty measure instead of constructing a new one.

We have modified it in the methods to state that this continuous frailty was constructed and verified by prior studies (page 4 in the revised manuscript):

“In addition, a previously established and verified continuous composite measure of frailty^{3,22} was also computed based on grip strength, gait speed, BMI, and fatigue.”

Section 2.5 Covariates: the rationale behind the selection of the covariates should be stated.

We agree with reviewer that it's important to clarify how these covariates were selected. Several demographics (age, sex, and education), sleep characteristics, as well as cardiovascular dysfunction have previously been reported to be associated with the risk of frailty. To test whether circadian rest-activity rhythms predict incident frailty independent of these known risk factors, we considered these covariates. We have added the rationale and corresponding literatures in the methods (Page 5 in the revised manuscript):

“Several demographics (age, sex, and education)^{1,23}, sleep characteristics²⁴, as well as cardiovascular dysfunction²⁵ are previously reported to be associated with the risk of frailty. To test whether circadian rest-activity rhythms predict incident frailty independent of these known risk factors, we considered the following covariates at baseline...”

Results

Lines 201-203: were predicted survival curves in Figure 2 based on the Cox model adjusting for baseline demographics?

Yes, the curves in the original Figure 2 were based on the Cox models adjusting for baseline demographics in the first version. Following the suggestion of reviewer 1 (see point 5 of Reviewer 1), we have revised Figure 2 in which the results from the fully adjusted models are shown.

Table 1: the measurement units of the variables should be included. The definition of ‘motor function’ is missing from the Methods.

We thank the reviewer for pointing out this issue. We have added the measurement units in Table 1. And we apologize for the confusion regarding motor function in Table 1. Because gait and grip (two frailty components) are based on the motor function tests, it makes little sense to consider ‘motor function’ as a separate variable in the analysis. We have fixed the error (removed motor function) in the revised Table 1.

Discussion

Lines 37-38: it would helpful to clarify the linkage between existing evidence on the sequence of emergence of frailty criteria and the study's finding of circadian rest-activity rhythms' association with specific frailty components (i.e., grip and BMI).

We thank the reviewer for bringing up this important point. The associations between circadian rest-activity rhythms and physical frailty appeared to be driven by the change in grip strength and BMI. While we further investigate prior evidence regarding which symptom(s) may emerge early among different components of frailty, we found that the observations vary significantly across studies. We have added these further explanations for our finding in the discussion (page 10 in the revised manuscript):

“Previous studies showed that frailty components may have different sequences of emergence before the onset of frailty that vary significantly across studies.^{38,39} For example, the results from the Women’s Health and Aging Study II (WHAS II)³⁹ showed that weakness (low grip strength) tended to develop first, while weight loss and exhaustion (fatigue) predicted more rapid development of frailty. Our findings of the associations between circadian rest-activity rhythms and grip strength, and BMI, may be related to the sequence in which each component occurs and its contribution to the development of frailty.”

References

1. Gao C, Li P, Morris CJ, et al. Actigraphy-Based Sleep Detection: Validation with Polysomnography and Comparison of Performance for Nighttime and Daytime Sleep During Simulated Shift Work. *Nat Sci Sleep*. 2022;14:1801-1816. doi:10.2147/NSS.S373107
2. Li P, Gao L, Gaba A, et al. Circadian disturbances in Alzheimer's disease progression: a prospective observational cohort study of community-based older adults. *The Lancet Healthy Longevity*. 2020;1(3):e96-e105. doi:10.1016/S2666-7568(20)30015-5
3. Gao L, Li P, Gaykova N, et al. Circadian Rest–Activity Rhythms, Delirium Risk, and Progression to Dementia. *Annals of Neurology*. 2023;93(6):1145-1157. doi:10.1002/ana.26617
4. Xiao Q, Qian J, Evans DS, et al. Cross-sectional and Prospective Associations of Rest-Activity Rhythms With Metabolic Markers and Type 2 Diabetes in Older Men. *Diabetes Care*. 2020;43(11):2702-2712. doi:10.2337/dc20-0557
5. Leng Y, Blackwell T, Cawthon PM, Ancoli-Israel S, Stone KL, Yaffe K. Association of Circadian Abnormalities in Older Adults With an Increased Risk of Developing Parkinson Disease. *JAMA Neurol*. 2020;77(10):1270-1278. doi:10.1001/jamaneurol.2020.1623
6. Hoogendijk EO, van Hout HPJ, Heymans MW, et al. Explaining the association between educational level and frailty in older adults: results from a 13-year longitudinal study in the Netherlands. *Annals of Epidemiology*. 2014;24(7):538-544.e2. doi:10.1016/j.annepidem.2014.05.002
7. Szanton SL, Seplaki CL, Thorpe RJ, Allen JK, Fried LP. Socioeconomic status is associated with frailty: the Women's Health and Aging Studies. *Journal of Epidemiology & Community Health*. 2010;64(01):63-67. doi:10.1136/jech.2008.078428
8. Buchman AS, Wilson RS, Bienias JL, Bennett DA. Change in frailty and risk of death in older persons. *Exp Aging Res*. 2009;35(1):61-82. doi:10.1080/03610730802545051
9. Buchman AS, Leurgans SE, Boyle PA, Schneider JA, Arnold SE, Bennett DA. Combinations of motor measures more strongly predict adverse health outcomes in old age: the rush memory and aging project, a community-based cohort study. *BMC Med*. 2011;9:42. doi:10.1186/1741-7015-9-42
10. Buchman AS, Boyle PA, Wilson RS, Tang Y, Bennett DA. Frailty is Associated With Incident Alzheimer's Disease and Cognitive Decline in the Elderly. *Psychosomatic Medicine*. 2007;69(5):483. doi:10.1097/psy.0b013e318068de1d
11. Boyle PA, Buchman AS, Wilson RS, Leurgans SE, Bennett DA. Physical frailty is associated with incident mild cognitive impairment in community-based older persons. *J Am Geriatr Soc*. 2010;58(2):248-255. doi:10.1111/j.1532-5415.2009.02671.x

Reviewers' Comments:

Reviewer #1:

Remarks to the Author:

I would like to thank the authors for their thoughtful response to my previous comments. The manuscript has much improved after the clarifications and the additional analysis. I don't have any further questions.

Reviewer #2:

Remarks to the Author:

The authors have answered all my queries. I have no further critiques.

Reviewer #3:

Remarks to the Author:

Thank you for addressing the comments. I have no further questions, but I would like to suggest that the manuscript could greatly benefit from a final proofread by a native English speaker.

Point-by-Point Response to Reviewers' Comments
Manuscript ID: NCOMMS-23-11633A
Title: Circadian disturbances and frailty risk in older adults

Reviewer #1:

I would like to thank the authors for their thoughtful response to my previous comments. The manuscript has much improved after the clarifications and the additional analysis. I don't have any further questions.

We sincerely thank the reviewer for their time and effort in reviewing our revised manuscript!

Reviewer #2:

The authors have answered all my queries. I have no further critiques.

We sincerely thank the reviewer for their time and effort in reviewing our revised manuscript!

Reviewer #3:

Thank you for addressing the comments. I have no further questions, but I would like to suggest that the manuscript could greatly benefit from a final proofread by a native English speaker.

We sincerely thank the reviewer for their time and effort in reviewing our revised manuscript! The uploaded version has been proof-read by our native English-speaking co-authors and colleagues.